# Comparative Semen Microbiota Composition of a Stallion in a *Taylorella equigenitalis* Carrier and Non-Carrier State

**DOI:** 10.3390/ani10050868

**Published:** 2020-05-17

**Authors:** Carlota Quiñones-Pérez, Amparo Martínez, Francisco Crespo, José Luis Vega-Pla

**Affiliations:** 1Laboratorio de Investigación Aplicada, Cría Caballar de las Fuerzas Armadas, Carretera de Madrid Km 395A, 14014 Córdoba, Spain; jvegpla@oc.mde.es; 2Genetics Department, University of Córdoba, edificio Gregor Mendel (C-5), Campus de Rabanales, 14071 Córdoba, Spain; amparomartinezuco@gmail.com; 3Centro Militar de Cría Caballar de Ávila, Cría Caballar de las Fuerzas Armadas, Calle Arsenio Gutiérrez Palacios, s/n, 05005 Ávila, Spain; fcrecas@oc.mde.es

**Keywords:** *Taylorella equigenitalis*, carrier, microbiome, stallion

## Abstract

**Simple Summary:**

Contagious equine metritis carriers have become a new cause of concern in horse stud farms. Their detection can result in significant financial loss and force owners to have their animals undergo antibiotic treatment. Current research has not been able to satisfactorily explain the appearance of carriers in agent-free farms. Studies made on microbial flora have given new insights into the diagnosis and treatment of different issues in animal systems. Next-generation sequencing (NGS) is a powerful tool that can draw an accurate picture of microbial flora. Therefore, the aim of this study was to compare the seminal bacterial composition of one stallion before and after being diagnosed with *Taylorella equigenitalis* using NGS. Our results show that the microbial seminal flora visibly changed between the samples analyzed. Corynebacteriaceae, an opportunistic bacterial family, was more common in the infected sample. However, Porphyromonadaceae, a natural component in several tissues, was more abundant in the negative sample. Despite the constraints of a single-case study, these findings can open the door to new therapeutic tools, as flora transplants. Similarly, seminal flora analysis may foresee microbial shifts, letting practitioners take preventive actions before a potential outbreak. Furthermore, these actions would have the extra benefit of reducing the administration of antibiotics to treat an infection.

**Abstract:**

Contagious equine metritis is receiving renewed attention due to the continuous detection of carriers in apparent agent-free farms. Interactions of *Taylorella* with the seminal microflora may be the plausible cause behind these spontaneous changes of the carrier state. Accordingly, the aim of this study was to compare the differences in the seminal microbiome composition of one stallion in the contagious equine metritis carrier state and non-carrier state. Samples were cryopreserved after their extraction. Cell disruption was performed by high-speed homogenization in grinding media. Bacterial families were identified via V3 amplification of the 16S rRNA gene and Ion Torrent sequencing. Only bacterial families with relative abundance above 5% were taken into consideration. The positive sample contained a strong dominance of Corynebacteriaceae (37.75%) and Peptoniphilaceae (28.56%). In the negative sample, the Porphyromonadaceae (20.51%), Bacteroidaceae (19.25%) and Peptoniphilaceae (18.57%) families prevailed. In conclusion, the microbiome seminal composition varies when an individual carries *Taylorella* from when it is free of it. The wider differences were found in the Corynebacteriaceae, Porphyromonadaceae and Bacteroidaceae families. Due to the limitations of a single-case analysis, further studies are needed for a better understanding of the stallion seminal microflora interactions.

## 1. Introduction

Contagious equine metritis (CEM) is a concerning condition in the horse industry, as its presence in livestock reduces fertility soundness in mares and can involve commercial restrictions. Its etiologic agent is *Taylorella equigenitalis*, a bacteria belonging to the Actinobacteria phylum. Stallions host the agent in the distal part of the urethra, becoming a long-timer carrier [1] if they are not submitted to a disinfection standard protocol [2,3].

Symptoms vary in stallions and mares. In mares, the disease manifests with endometritis, cervicitis and vaginitis of variable severity, and it sometimes appears a mucopurulent vaginal discharge. It usually leads to temporary infertility. The recovery is uneventful, but the animal becomes a carrier. There are no symptoms in stallions, but they also become carriers. Official diagnostic methods includes culture and detection of the agent by PCR from swabs taken from predilection sites [4].

Recently, attention has been brought to CEM, as new carriers have been detected in non-symptomatic farms [5]. This has raised questions about the transmission mechanisms of the bacteria [6] or even the effectiveness of diagnostic methods [7]. It has been stated that official methods may not detect low microbial concentration carriers [8,9] or that the nature of this agent makes it difficult to isolate it in culture-based methods [4]. However, next-generation sequencing studies performed in humans have shown that certain seminal microbiome compositions could lower the chance of venereal agents of surviving [10,11,12,13].

Our study presents the case of the seminal microbiome shift in a horse that underwent a spontaneous reversion from a *Taylorella equigenitalis* carrier state to a non-carrier one. Therefore, the aim of this study was to compare the differences in the seminal microbiome composition of one stallion in its *Taylorella equigenitalis* carrier and non-carrier state.

## 2. Materials and Methods

### 2.1. Sample Collection

The sample collection was performed in the Equine Breeding Centre of the Spanish Army located in Écija under the Chief of Unit authorisation. Animals are handled in accordance with Spanish law for animal welfare (Law 32/07). Animals were not submitted to extra semen extractions for our experiment sample collection nor was their daily workflow interrupted. Samples were not collected for the purpose of the study.

Samples were collected in two different batches. The first batch was composed of semen samples from six stallions. Age ranged from 8 to 18 years old. Samples were collected using an in-line gel-filter Missouri artificial vagina, with mare in oestrus as a teaser. An inner disposable plastic liner was used with each animal to avoid cross-contamination. Animals lived under the same dietary and exercise conditions, with no clinical diseases reported. Official analysis for contagious equine metritis tests [4] were negative in all cases, so animals did not receive any type of treatment. Sperm quality tests were also performed prior to seminal extraction (total volume, gel-free volume, sperm concentration, total motility and progressive motility).

Samples were cryopreserved immediately after their extraction. All samples were then analysed using next-generation sequencing following the process described below. After a beta diversity study, we found out that one of the samples was manifestly different.

This finding led to the collection of a second batch six months afterwards. It was composed of semen samples from 14 different stallions, including the animals from the first batch. Age ranged from 6 to 18 years old in this case. The collection process, environmental conditions and analysing methods post-collection were the same as the ones for the first batch.

### 2.2. Control Sample

For each of the batches, a pattern sample (ZymoBIOMICS Microbial Community Standard^®^, Zymo Research, Irvine, CA, USA) was included in order to evaluate the quality of the DNA extraction and its amplification.

### 2.3. DNA Extraction

DNA extraction was performed using a ZymoBIOMICS^®^ DNA Miniprep kit (Zymo Research, Irvine, CA, USA) commercial kit. Samples were previously submitted to a combination of mechanic and enzymatic-digestion cell disruption as described by Bag [14]. Then, DNA extraction was performed following the manufacturer’s instructions.

### 2.4. Next-Generation Sequencing Analysis

Amplicons were obtained using an Ion 16S Metagenomics^®^ kit (Thermo Fisher, Waltham, MA, USA). This kit characterizes five different sets of 16S hypervariable regions, V2, V3, V4, V67 and V8. The library was constructed with an Ion Plus Fragment Library kit and amplicons were labelled with an Xpress™ Barcode Adapters 1–16 kit. Samples were then pooled using Ion PGM^®^ (Thermo Fisher, Waltham, MA, USA), HiQ Sequencing kit^®^ (Thermo Fisher, Waltham, MA, USA), Ion 316 v2 BC^®^ chip (Thermo Fisher, Waltham, MA, USA) and sequenced using the Ion 16S™ Metagenomics Workflow in Ion Reporter™ Software (Thermo Fisher, Waltham, MA, USA).

Data analysis was performed in the Ion Reporter server system (https://ionreporter.thermofisher.com/ir/secure/home.html). β-diversity was calculated using Bray-Curtis dissimilarity analysis. Hypervariable region V3 was chosen for taxonomic bacterial identification, as it has been suggested to detect a wider range of bacterial species [15].

### 2.5. Statistical Analysis

Similarity of samples was compared calculating the Bray-Curtis dissimilarity index according to the following formula:$${\mathrm{BC}}_{a,b}=1-\frac{2C_{a,b}}{S_{a}+S_{b}}$$
where a and b are two samples; S_a_ is the total number of specimens counted on site a; S_b_ is the total number of specimens counted on site b; and C_a,b_ is the sum of the lesser counts of each bacterial family found in both sites. Value of 0 means the two samples have the same composition, and 1 means the two samples do not share any species.

Bray-Curtis indexes were then compared using Pearson’s χ^2^ test. Significant values were consider when *p* < 0.1.

## 3. Results

The resulting composition of the quality control had a minor variation (±5.6% maximum) comparing to the composition provided by the manufacturer.

### 3.1. Microbial Community Structure Differences Between Batches

Beta diversity analysis was used to compare microbial community structure between seminal samples of different stallions at a sampling point [16]. In the first sample batch, we observed one outlier within the group (Figure 1).

We calculated the Bray-Curtis (B-C) dissimilarity index to determine the degree of similarity among samples. Significant differences were found between the *Taylorella equigenitalis* carrier indexes and the rest of indexes (*p* < 0.1). Results are represented in Table 1.

This sample was reanalysed six months later following the same method within a second batch of 14 samples. This time, the former outlier sample belonged to the main group (Figure 2). Then, we also calculated the B-C dissimilarity index for this batch (Table 2). This time, no significant differences were found between samples (*p* > 0.1).

### 3.2. Taxonomic Composition of Seminal Samples

A general taxonomic composition of batches 1 and 2 is represented in Figure 3. The most abundant phyla were Firmicutes and Bacteroidetes in all cases, except for the *Taylorella equigenitalis* carrier (TE+). The two most dominant phyla for TE+ were Firmicutes and Actinobacteria.

Mean values of batch 1 were 36.33% for Firmicutes phylum, 37.33% for Bactoroidetes; 15.33% for Actinobacteria; 5.50% for Proteobacteria; 1.50% for Fusobacteria; 2.00% for Spirochaetes; and 2.23% for Synergestes. Mean values of batch 2 were 29.50% for Firmicutes phylum, 45.86% for Bactoroidetes; 13.71% for Actinobacteria; 5.27% for Proteobacteria; 1.86% for Fusobacteria; 2.91% for Spirochaetes; and 0.89% for Synergestes.

### 3.3. Comparative Taxonomic Composition of TE+ and TE−

Figure 4 shows relative abundance at the family level in samples TE+ and TE− (stallion cleared of *Taylorella equigenitalis*). Only common families with a relative abundance above 5% are represented.

Taxonomic composition classification revealed that TE+ contained a 0.02% of the Alcaligenaceae family (represented by *Taylorella equigenitalis* species), while TE− showed no remainders of the *Taylorella equigenitalis* species nor its complete family.

As far as general bacterial composition is concerned, TE+ presents a microbial diversity of 32 families, with a strong dominance of Corynebacteriaceae (up to 37.75%) and Peptoniphilaceae (28.56%). The percentage of the following most common bacteria, Ruminococcaceae, only reaches 6.82%. Regarding TE−, it contains 31 different bacteria families, with a dominance of Porphyromonadaceae (20.51%), Bacteroidaceae (19.25%) and Peptoniphilaceae (18.57%). Corynebacteriaceae is the fourth most abundant family with a 10.64% presence. The family composition of TE+ also contrasts with the composition of the rest of samples. Thus, mean values for the families are represented in Table 3.

## 4. Discussion

Our results show there is a different microbiome composition when an animal carries *Taylorella equigenitalis* than when it is free of the agent. It has already been proven that some microflora changes in the equine digestive tract can favor the growth of some pathogens [17,18,19]. In the genital tract, it has also been hypothesized that infections or strange agents may produce changes in the environment around them and, therefore, favor some bacteria families to grow while hindering others [20].

In our case, Actinobacteria phylum highly differs between TE+ and TE− (41% vs. 14%, respectively). The main family of this phylum, Corynebacteriaceae, has been regularly described as a normal component of seminal flora in humans [20,21,22,23,24,25] as well as in stallions [26,27,28,29,30,31,32]. Meanwhile, other authors maintain that it holds an opportunistic nature, and even some have related its presence with a higher caspases activity [33].

Concerning Bacteroidetes phylum, the three most common families have largely varied from the carrier situation to the non-carrier one. Mändar have already pointed out that the presence of Prevotellaceae family alone or combined with Porphyromonadaceae are associated with a higher rate of reproductive inflammatory conditions [23]. Our results may contradict this finding, as they are the predominant group in the non-carrier state, especially Porphyromonadaceae. It is relevant to say that it is not easy to compare the effect of the Bacteroidetes phylum with previous studies, as it is a difficult to culture group and most studies utilized cultured-based methods.

In Mändar [23], it was possible to divide a population in three groups according to their seminal microbiome composition, that time relating the results with their seminal quality. In our case, the division would create two groups: the Corynebacteriaceae family predominant group, representing the contagious equine metritis carriers; and the Bacteroidetes phylum predominant group for the non-carrier stallions. It would be of the utmost importance to dig deeper into the mechanisms underlying these differences, as understanding the seminal microbiome composition can open the door to a future diagnostic or even prophylactic tools.

Finally, it is necessary to indicate some strengths and limitations of this study. Next-generation sequencing overcomes limitations of culture-based diagnostic methods. It can detect fastidious-to-cultivate genera and it is less affected by contamination [34]. In spite of that, NGS also has limitations. For example, differential amplification of primers [35], DNA extraction or interpretation of data [12]. Another limitation of our study would be sample size.

## 5. Conclusions

The microbiome seminal composition varies when an individual carries *Taylorella equigenitalis* from when it is free of the agent. The wider differences were found in the Corynebacteriaceae family, increased in the carrier case; and the Porphyromonadaceae and Bacteroidaceae families, increased in the non-carrier case. Besides, we have observed it is possible to detect CEM carriers using NGS. However, due to the limitations of a one-subject case report, further studies are needed in order to completely comprehend the interactions that occur in the stallion seminal microflora.

## Figures and Tables

**Figure 1 animals-10-00868-f001:**
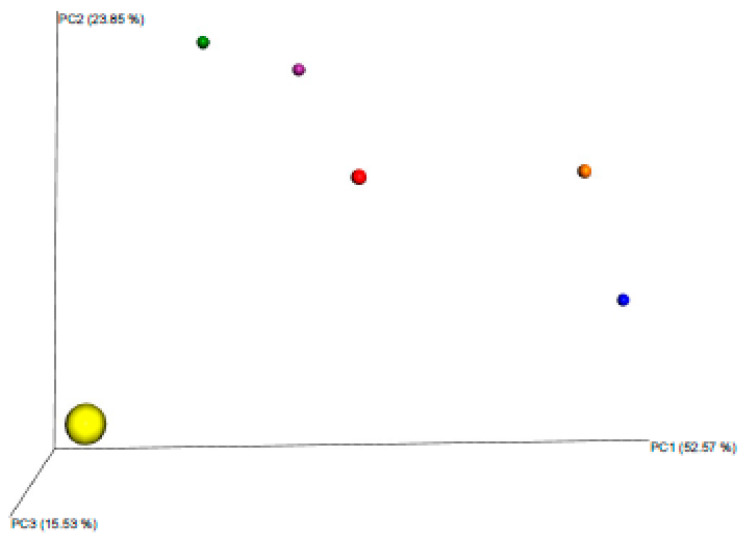
Family β-diversity using Bray-Curtis dissimilarity analysis of the first six-sample batch. There is a group of five samples in the upper-right area of the graph, while one sample is isolated on the down-left corner. The highlighted dot corresponds to *T equigenitalis* positive sample. Significant differences were found between the *Taylorella equigenitalis* carrier composition and the rest of samples (*p* < 0.1).

**Figure 2 animals-10-00868-f002:**
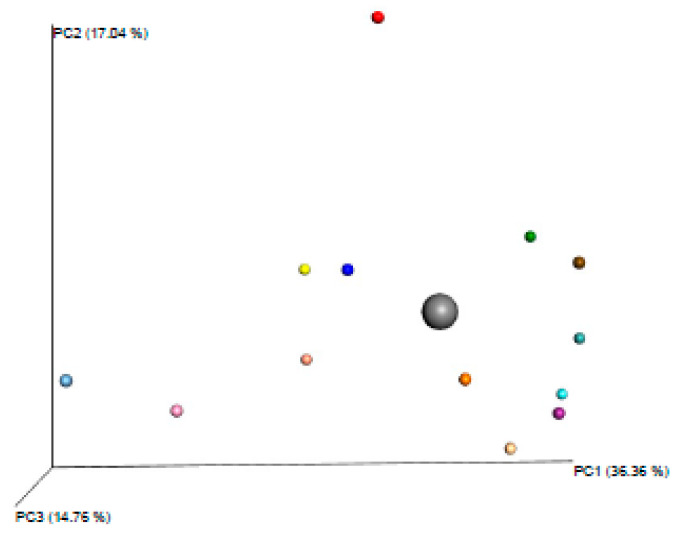
Family β-diversity using Bray-Curtis dissimilarity analysis of the second 14-sample batch. The highlighted dot corresponds to *T equigenitalis* negative sample. No significant differences were found between the *Taylorella equigenitalis* negative sample composition and the rest of samples (*p* > 0.1).

**Figure 3 animals-10-00868-f003:**
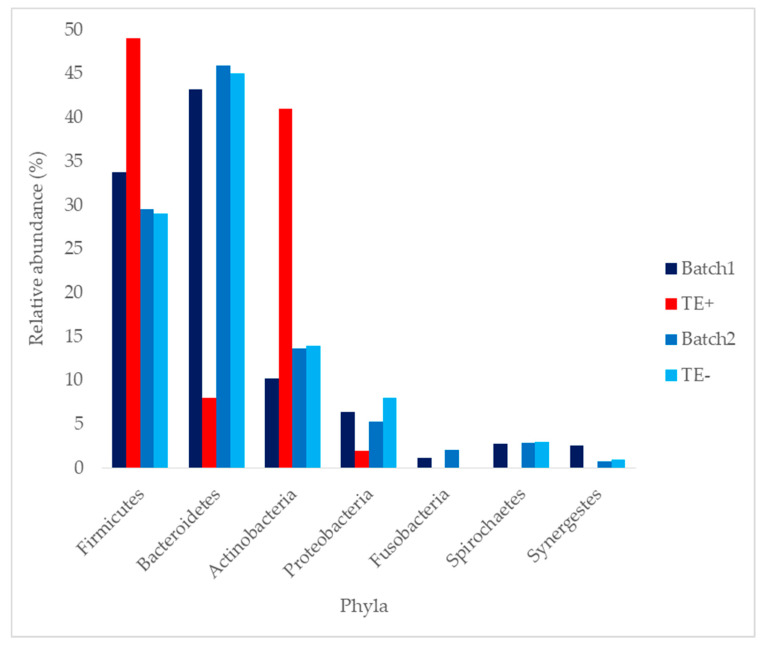
Mean phylum taxonomic composition of batch 1, TE+, batch 2 and TE−. It is observed how TE+ phylum composition highly varies compared with batch 1, batch 2 and TE−. TE+: *Taylorella equigenitalis* carrier. TE−: stallion cleared of *Taylorella equigenitalis*. Results are expressed as percentage (%).

**Figure 4 animals-10-00868-f004:**
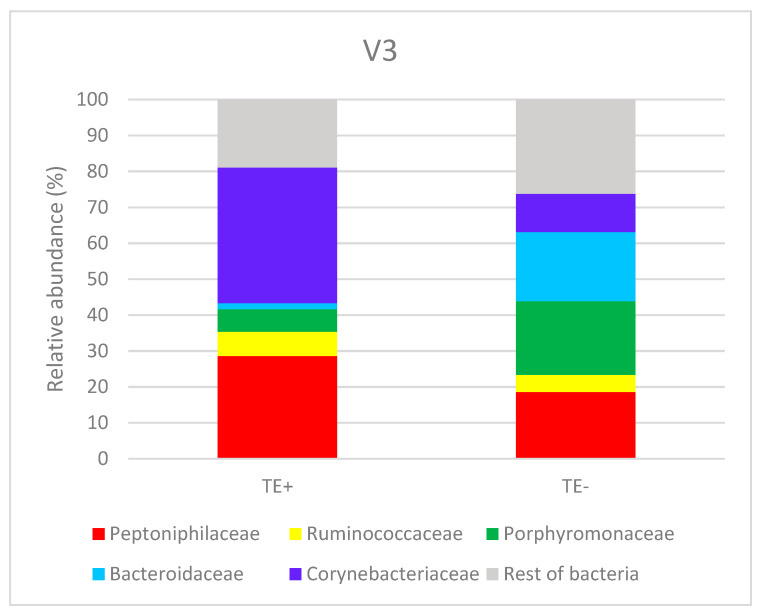
Comparative taxonomic composition of TE+ and TE−. The results are expressed as percentages (%). TE+ corresponds to *T equigenitalis* positive sample and TE− to the stallion cleared of *T equigenitalis*. Only common families with a relative abundance above 5% are included separately.

**Table 1 animals-10-00868-t001:** Bray-Curtis dissimilarity indexes for the first batch.

Stallions	1	2	3	4	5	TE+
1	-	-	-	-	-	-
2	0.40	-	-	-	-	-
3	0.57	0.37	-	-	-	-
4	0.51	0.55	0.66	-	-	-
5	0.53	0.51	0.63	0.37	-	-
TE+ *	0.64	0.66	0.64	0.78	0.58	-

Values near 0 means the two samples have the same composition. Values near 1 means the two samples do not share any species. Numbers represent animals. TE+: *Taylorella equigenitalis* carrier. * group of samples with *p* < 0.1.

**Table 2 animals-10-00868-t002:** Bray-Curtis dissimilarity indexes for the second batch.

Stallions	1	2	3	4	5	6	7	8	9	10	11	12	13	TE−
1	-	-	-	-	-	-	-	-	-	-	-	-	-	-
2	0.39	-	-	-	-	-	-	-	-	-	-	-	-	-
3	0.57	0.37	-	-	-	-	-	-	-	-	-	-	-	-
4	0.51	0.54	0.65	-	-	-	-	-	-	-	-	-	-	-
5	0.54	0.51	0.61	0.38	-	-	-	-	-	-	-	-	-	-
6	0.49	0.54	0.62	0.44	0.25	-	-	-	-	-	-	-	-	-
7	0.53	0.42	0.42	0.67	0.54	0.51	-	-	-	-	-	-	-	-
8	0.54	0.48	0.59	0.38	0.22	0.12	0.53	-	-	-	-	-	-	-
9	0.45	0.47	0.51	0.33	0.15	0.29	0.44	0.34	-	-	-	-	-	-
10	0.60	0.60	0.70	0.14	0.30	0.39	0.68	0.32	0.47	-	-	-	-	-
11	0.52	0.53	0.58	0.49	0.36	0.30	0.52	0.28	0.34	0.42	-	-	-	-
12	0.53	0.39	0.48	0.60	0.47	0.55	0.54	0.55	0.49	0.60	0.54	-	-	-
13	0.45	0.34	0.50	0.61	0.47	0.50	0.34	0.53	0.37	0.61	0.51	0.47	-	-
TE−	0.46	0.46	0.53	0.65	0.37	0.45	0.48	0.45	0.30	0.56	0.34	0.47	0.38	-

Values near 0 means the two samples have the same composition. Values near 1 means the two samples do not share any species. Numbers represent animals. TE−: stallion cleared of *Taylorella equigenitalis*.

**Table 3 animals-10-00868-t003:** Mean values for Porphyromonadaceae, Bacteroidaceae, Prevotellaceae and Corynebacteriaceae families of the TE+ sample and the first batch of samples without TE+.

Bacterial families	TE+	Rest of Samples
Porphyromonadaceae	6.24	30.07 ± 16.18
Bacteroidaceae	1.68	0.80 ± 0.60
Prevotellaceae	0.08	10.72 ± 6.43
Corynebacteriaceae	37.75	7.87 ± 2.38

Results are expressed as percentage (%) and mean ± standard error of the mean. TE+ corresponds to *Taylorella equigenitalis* positive sample.

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
