# Peer review of "Comparative Semen Microbiota Composition of a Stallion in a Taylorella equigenitalis Carrier and Non-Carrier State"

_animals, 2020, doi:10.3390/ani10050868_

Round 1

Reviewer 1 Report

Comments to the authors:

The manuscript describes the changes in seminal microbiome of a stallion that was diagnosed based on NGS as Taylorella equigenitallis carrier and 6 months later cleared off this pathogen.

Was Taylorella equigenitallis identified by taxonomic composition at genus (and species) level, or assumed only by Alcaligenaceae family presence? This family includes other genus such as Bordetella.

The Materials and Methods section is much improved. It is not necessary to divide in two sections 2.1. materials and 2.2 Methods, the subheadings are enough.

Did you find statistical significance between the community structure of TE+ stallion and the other stallions with your beta diversity analysis? Or was only a difference that was observed visually in the PC graph?

Did you calculate other Beta-diversity metrics (e.g. bray-curtis, jaccard, unifrac), which are more appropriate for clustering than Euclidean distances? If so, please report in material and methods section. If results were not statistically significant then report so.

Please divide Results section in subheadings which describe the main findings represented in the figures and tables as follows:

  • Beta diversity
    • Figure 1 and 2 representing microbial community structure differences
  • Taxonomic composition of seminal samples in different stallions
    • Need graph or table showing differences in most abundant phylum, family and genus composition between non-carrier stallions and carrier stallion at sampling point 1 (batch 1) and another graph between non-carrier stallion and the cleared stallion at sampling point 2 (batch 2)
  • Taxonomic composition of seminal samples in a TE carrier stallion
    • Need graph or table showing only results from taxonomic composition of stallion described displaying sample 1 (TE+) and sample 2 (TE-). If Figure 3 is this what is displaying change text and legend so that can be understood
    • Need to show most abundant phylum and genus composition in addition to family composition

Adapt discussion to the results (i.e. discuss beta diversity findings and then phylum, family and genus differences). Include a paragraph on limitations of study design to draw such conclusions in discussion section.

Specific comments:

Line 56: substitute “made” by “performed”

Line 57-58: Rephrase “could hinder the chances of venereal agents surviving”

Line 66. Remove “The”

Line 68 Substitute “raised” by “handled”

Line 69 Add extra “semen” extractions

Line 73 Change to “mare in oestrus”

Line 73 Remove “so”

Line 82 Change to “semen samples from 14 different stallions” Clarify if the 6 stallions from the first batch were all part of the second batch.

Line 104: “Taxonomic” bacterial identification

Line 109: “had” instead of “has”

Line 107 to 110: Thanks for the clarification on quality control. This paragraph should be included in Material and Methods instead of Results). Table 1 is not necessary.

Line 115 “was” instead of “is”. Please correct verbs in all manuscript verb to past tense.

Line 115-116 Beta diversity analysis was used to compare microbial community structure between seminal samples of different stallions at a sampling point. In the first sample batch, …

Line 116 Specify if the outlier was statistically significantly different form the other stallions. Correct this nomenclature in the figure 1 legend

Line 122 Rather than saying that the former outlier belong to the group, state if the previously TE+ seminal sample was not statistically significant different from the rest of the samples at this sampling point. Correct this nomenclature at the figure 2

Line 128 Remove “a closer look…” Report as “Taxonomic composition classification revealed…” Clarify if Taylorella equigenitallis was identified by taxonomic composition at genus (and species) level and include its corresponding relative abundance percentage.

Line 137 Change to “Relative abundance at the family level”

Figure 3 Add unit of measure “Relative abundance (%)” in the Y axis

Line 159-165 This belongs to results section.

Line 166-168 In which samples and which species?

Author Response

Comments to the authors:

The manuscript describes the changes in seminal microbiome of a stallion that was diagnosed based on NGS as Taylorella equigenitallis carrier and 6 months later cleared off this pathogen.

Was Taylorella equigenitallis identified by taxonomic composition at genus (and species) level, or assumed only by Alcaligenaceae family presence? This family includes other genus such as Bordetella. Yes, it was identified at species level as Taylorella equigenitalis. Besides, we found no other Alcaligenaceae species in the sample.

The Materials and Methods section is much improved. It is not necessary to divide in two sections 2.1. materials and 2.2 Methods, the subheadings are enough. OK, these subsections have been removed. We have added a new subsection: statistical analysis.

Did you find statistical significance between the community structure of TE+ stallion and the other stallions with your beta diversity analysis? Or was only a difference that was observed visually in the PC graph? Firstly, it was a difference we observed in the PC graph. Then, after this first revision, we calculated Bray-Curtis indexes and we compared them using Pearson’s χ2. We found significant differences (p<0.1). This new information is now on the Results section (in lines 132-150).

Did you calculate other Beta-diversity metrics (e.g. bray-curtis, jaccard, unifrac), which are more appropriate for clustering than Euclidean distances? If so, please report in material and methods section. If results were not statistically significant then report so. Yes, we also calculated the Bray-Curtis dissimilarity index at family level. We firstly decided to include the Euclidean distances graphs because PC axes had higher values. In spite of that, we have revised some literature [1–3] and you are right: the vast majority of the authors think it is more appropriate to use a Bray-Curtis analysis. Therefore, we have removed the Euclidean test and included the B-C. As we pointed in the previous comment, we also included some statistic treatment. New data is in the Results section.

Please divide Results section in subheadings which describe the main findings represented in the figures and tables as follows:

Beta diversity Figure 1 and 2 representing microbial community structure differences. This subheading is now in line 123 and it is called Microbial community structure differences between batches. Figures 1 and 2 have changed (Euclidean for Bray-Curtis) and we added two tables (B-C indexes).

Taxonomic composition of seminal samples in different stallions Need graph or table showing differences in most abundant phylum, family and genus composition between non-carrier stallions and carrier stallion at sampling point 1 (batch 1) and another graph between non-carrier stallion and the cleared stallion at sampling point 2 (batch 2). We have created a new table (Table 1) including the phyla composition of all samples. We also added a brief description of the phyla founded and used some basic statistic (means) to compare TE+ with the rest of samples. We have preferred not adding the family/genus composition of non-carriers because we would like to focus just on the converted stallion. This subheading starts in line 152 and it is called taxonomic composition of seminal samples.

Taxonomic composition of seminal samples in a TE carrier stallion Need graph or table showing only results from taxonomic composition of stallion described displaying sample 1 (TE+) and sample 2 (TE-). If Figure 3 is this what is displaying change text and legend so that can be understood. Yes, it was Figure 3. We have changed the legend to avoid confusion. We have created a new subheading (line 165), which it is now called Comparative taxonomic composition of TE+ and TE-.

Adapt discussion to the results (i.e. discuss beta diversity findings and then phylum, family and genus differences). After considering your comment, we have finally decided to structure discussion around the family level. NGS analysis can only reach to the genus level in certain species (as Taylorella). It usually groups similar bacteria in subfamilies and sometimes genus, but it is not very precise at that level. Include a paragraph on limitations of study design to draw such conclusions in discussion section. A paragraph discussing the limitations of our study has been included (lines 212-216).

Specific comments:

Line 56: substitute “made” by “performed”. Done.

Line 57-58: Rephrase “could hinder the chances of venereal agents surviving”. Done

Line 66. Remove “The”. Done

Line 68 Substitute “raised” by “handled”. Done

Line 69 Add extra “semen” extractions. Done

Line 73 Change to “mare in oestrus”. Done

Line 73 Remove “so”. Done

Line 82 Change to “semen samples from 14 different stallions” Done. Clarify if the 6 stallions from the first batch were all part of the second batch. Yes, the first six semen samples were included in batch 2. This clarification is now in lines 87-88.

Line 104: “Taxonomic” bacterial identification. Done

Line 109: “had” instead of “has”. Done

Line 107 to 110: Thanks for the clarification on quality control. This paragraph should be included in Material and Methods instead of Results) We have included the control information in the Material and Methods section (lines 90-93). Table 1 is not necessary. We have removed Table 1. We have just stated the percentage of variation between the composition of the quality control and our results. If that is not representative enough, we could reintroduce the Table.

Line 115 “was” instead of “is”. Done. Please correct verbs in all manuscript verb to past tense. Revision done.

Line 115-116 Beta diversity analysis was used to compare microbial community structure between seminal samples of different stallions at a sampling point. In the first sample batch, … Construction changed.

Line 116 Specify if the outlier was statistically significantly different form the other stallions. Correct this nomenclature in the figure 1 legend. As you previously suggested, we have changed the statistical treatment from Euclidean distances to Bray-Curtis dissimilarity analysis. We have calculated the B-C dissimilarity index in both batches to make a statistical comparison. As it now appears in the M&M section, we used the Pearson’s χ2 test to assess significant differences between TE+ indexes and the rest of samples indexes. In batch 1, we found significant differences considering p<0.1.

Line 122 Rather than saying that the former outlier belong to the group, state if the previously TE+ seminal sample was not statistically significant different from the rest of the samples at this sampling point. Correct this nomenclature at the figure 2. For batch 2, we applied the same statistical treatment as for batch 1. This time, we found no significant differences between TE- indexes and the rest of samples indexes (p<0.1). We really hope this new treatment gives sufficient statistical power to our study.

Line 128 Remove “a closer look…” Report as “Taxonomic composition classification revealed…” Done. Clarify if Taylorella equigenitallis was identified by taxonomic composition at genus (and species) level and include its corresponding relative abundance percentage. It was identified by taxonomic composition, with a relative abundance of 0.02% (information included in lines 172-173).

Line 137 Change to “Relative abundance at the family level”. Done

Figure 3 Add unit of measure “Relative abundance (%)” in the Y axis. Done

Line 159-165 This belongs to results section. Done

Line 166-168 In which samples and which species? This study was made in humans, as it is the species where seminal microbiome has been studied the most. In animals, to the best of our knowledge, there are only a few seminal microbiome studies in mice.

REFERENCES:

  1. Clarke, K.R.; Somerfield, P.J.; Chapman, M.G. On resemblance measures for ecological studies, including taxonomic dissimilarities and a zero-adjusted Bray–Curtis coefficient for denuded assemblages. Journal of Experimental Marine Biology and Ecology 2006, 330, 55–80, doi:10.1016/j.jembe.2005.12.017.
  2. Pavoine, S.; Ricotta, C. Measuring functional dissimilarity among plots: Adapting old methods to new questions. Ecological Indicators 2019, 97, 67–72, doi:10.1016/j.ecolind.2018.09.048.
  3. Ricotta, C.; Podani, J. On some properties of the Bray-Curtis dissimilarity and their ecological meaning. Ecological Complexity 2017, 31, 201–205, doi:10.1016/j.ecocom.2017.07.003.

Reviewer 2 Report

Dear Authors,

How I understand:

(CEM) is a disease of the mucous membranes of the reproductive system induced by Taylorella equigenitalis. The spread of CEM may cause significant economic losses resulting from the periodic mare's infertility, early pregnancy loss and embryo mortality, and also limiting horse turnover and/or trade-in the semen of stallions. New carriers have been detected in non-symptomatic farms and thus This has raised questions about the transmission mechanisms of the bacteria or even the effectiveness of diagnostic methods.

However, the manuscript needs support with writing and editorial.

Introduction:

I cannot find the description of Taylorella equigenitalis infection in equines.

Some information like that should be in:” Infectious equine metritis (Contagious Equine Metritis CEM) is a disease of the mucous membranes of the reproductive system induced by Taylorell equigenitalis. The condition manifests with mucopurulent discharge from the uterus and various degrees of cervicitis and vagina, followed by temporary infertility for several weeks. The infected animal becomes a carrier. Stallions do not have symptoms and become an asymptomatic bacterial carrier. Routine diagnosis of the disease is based on bacteriological examination of swabs taken from predilection sites.” 

Ln 59-60: this has to be far better describe with literature citation about this spontaneous reversion

Material and methods

Animals and sample collection:

Animals: please write becomes about stallions and study design. Does the group contain young or old stallions? Do you have information about breeding performance, the fertility of stallions? How was breeding performed by insemination or natural covering??

Ln 69: Instead of writing that you were not collecting samples for the purpose of this study please explain the purpose of sample collection. Was this before breading season routine check? Or in the mid-seasonal check?

Ln 116 and 128-136

Was this diagnosis? Why is that? Treatment?

Author Response

Dear Authors,

How I understand:

(CEM) is a disease of the mucous membranes of the reproductive system induced by Taylorella equigenitalis. The spread of CEM may cause significant economic losses resulting from the periodic mare's infertility, early pregnancy loss and embryo mortality, and also limiting horse turnover and/or trade-in the semen of stallions. New carriers have been detected in non-symptomatic farms and thus This has raised questions about the transmission mechanisms of the bacteria or even the effectiveness of diagnostic methods.

However, the manuscript needs support with writing and editorial.

Introduction:

I cannot find the description of Taylorella equigenitalis infection in equines. Some information like that should be in:” Infectious equine metritis (Contagious Equine Metritis CEM) is a disease of the mucous membranes of the reproductive system induced by Taylorella equigenitalis. The condition manifests with mucopurulent discharge from the uterus and various degrees of cervicitis and vagina, followed by temporary infertility for several weeks. The infected animal becomes a carrier. Stallions do not have symptoms and become an asymptomatic bacterial carrier. Routine diagnosis of the disease is based on bacteriological examination of swabs taken from predilection sites.” OK, we added a brief description of CEM infection in lines 52-56 (Introduction section).

Ln 59-60: this has to be far better describe with literature citation about this spontaneous reversion. This is one of the key points of our study. In the last few years, there has been an upturn of CEM carriers in breeding centres. Many of these new cases have no epidemiological explanation (no use of natural covering, good hygienic practices and so on). Some authors have pointed to a low sensitivity of official diagnostic methods (1–3). This would produce inconsistencies in the results of CEM monitoring plans. However, official diagnostic methods, culture and PCR (4), have undergone few variations since its first use in the late 70’s and late 90’s, respectively. Then, this hypothesis could not explain the current raise of cases.

With our study, we wanted to open the door to a new, plausible explanation of this upturn of cases: natural microflora dwelling in genital tract can regulate the presence of pathogenic/opportunistic bacteria. Under this hypothesis, there could be stallions with a low or no charge of Taylorella equigenitalis (not detectable by official methods) that could be upregulated under certain conditions. This would lead to a positive CEM result in following test, no changes in management needed. This also applies the other way round: a former positive animal can draw a future negative result (no treatment involved) due to a microflora regulation. These microflora interactions has been previously described in some conditions in the genital tract of men (5–7); and also in the digestive system of horses (8–12).

For sure, a single-case study can only draw some preliminary results. Our hypothesis needs a much deeper examination. In fact, we are currently working on another study with a larger number of stallions trying to understand seminal microflora interactions.

Material and methods

Animals and sample collection:

Animals: please write becomes about stallions and study design. Does the group contain young or old stallions? Yes, age range goes from 6 to 18 years old. This information is now included in lines 76 and 88. Do you have information about breeding performance, the fertility of stallions? Yes, all stallions had good results in sperm quality tests. Besides, all of them have drawn good fertility results in mares from the same breeding centre and in animals from customers. Unfortunately, we cannot give an exact pregnancy rate or fertility percentage. How was breeding performed by insemination or natural covering?? Breeding is always performed by artificial insemination (information included in lines 76-78).

Ln 69: Instead of writing that you were not collecting samples for the purpose of this study please explain the purpose of sample collection. Was this before breading season routine check? Or in the mid-seasonal check? OK, for a better understanding of the extraction process, we made some changes in the paragraph from lines 77 to 83. Concerning the purpose of the collection, the breeding centre receives customer’s request all along the year, except from July to mid-September. Our work was performed during breeding season, so semen collection was part of the centre daily routine. As part of their protocol, after each extraction, semen is always submitted to a sperm quality test. It includes volume, gel-free volume, sperm concentration, total motility and progressive motility (data not included in the manuscript).

Ln 116 and 128-136

Was this diagnosis? Why is that? Yes, official diagnosis (culture) did not detect the presence of the bacteria in either of the cases. We detected the presence of Taylorella equigenitalis in one of the samples with NGS. These CEM false negatives have already been described in some studies (13,14). Even the World Organization for Animal Health points that the bacteria is difficult to isolate (4), so culture may not always detected positive cases. With high-throughput sequencing, it is possible to classify bacteria to species level, without the limitations of culture (15). Treatment? As official tests were negative, veterinarians were not aware of the latent infection. Consequently, the animal did not receive any treatment (information included in lines 82-83).

REFERENCES.

  1. Kristula M, Smith B. Diagnosis and treatment of four stallions, carriers of the contagious metritis organism--case report. Theriogenology. 2004 Jan;15(61):2–3.
  2. Schulman ML, May CE, Keys B, Guthrie AJ. Contagious equine metritis: Artificial reproduction changes the epidemiologic paradigm. Veterinary Microbiology. 2013 Nov 29;167(1):2–8.
  3. Timoney PJ. Horse species symposium: contagious equine metritis: an insidious threat to the horse breeding industry in the United States. J Anim Sci. 2011 May;89(5):1552–60.
  4. World Organisation for Animal Health. Manual of diagnostic tests and vaccines of terrestrial animals. 2018.
  5. Liu CM, Osborne BJW, Hungate BA, Shahabi K, Huibner S, Lester R, et al. The semen microbiome and its relationship with local immunology and viral load in HIV infection. PLoS Pathog. 2014 Jul;10(7):e1004262.
  6. Hou D, Zhou X, Zhong X, Settles ML, Herring J, Wang L, et al. Microbiota of the seminal fluid from healthy and infertile men. Fertil Steril. 2013 Nov;100(5):1261–9.
  7. Weng S-L, Chiu C-M, Lin F-M, Huang W-C, Liang C, Yang T, et al. Bacterial Communities in Semen from Men of Infertile Couples: Metagenomic Sequencing Reveals Relationships of Seminal Microbiota to Semen Quality. Abdo Z, editor. PLoS ONE. 2014 Oct 23;9(10):e110152.
  8. Niu Q, Zhang L, Zhang K, Huang X, Hui F, Kan Y, et al. Changes in intestinal microflora of Caenorhabditis elegans following Bacillus nematocida B16 infection. Sci Rep. 2016 Feb 2;6:20178.
  9. Morrison PK, Newbold CJ, Jones E, Worgan HJ, Grove-White DH, Dugdale AH, et al. The Equine Gastrointestinal Microbiome: Impacts of Age and Obesity. Front Microbiol. 2018;9:3017.
  10. Clark A, Sallé G, Ballan V, Reigner F, Meynadier A, Cortet J, et al. Strongyle Infection and Gut Microbiota: Profiling of Resistant and Susceptible Horses Over a Grazing Season. Front Physiol. 2018;9:272.
  11. Salem SE, Maddox TW, Antczak P, Ketley JM, Williams NJ, Archer DC. Acute changes in the colonic microbiota are associated with large intestinal forms of surgical colic. BMC Vet Res. 2019 Dec 21;15(1):468.
  12. Biddle AS, Tomb J-F, Fan Z. Microbiome and Blood Analyte Differences Point to Community and Metabolic Signatures in Lean and Obese Horses. Front Vet Sci. 2018;5:225.
  13. Aznai T, Wada R, Okuda T, Aoki T. Evaluation of the field application of PCR in the eradication of contagious equine metritis from Japan. - PubMed - NCBI. J Vet Med Sci. 2002;11(64):999–1002.
  14. Nadin-Davis S, Knowles MK, Burke T, Böse R, Devenish J. Comparison of culture versus quantitative real-time polymerase chain reaction for the detection of Taylorella equigenitalis in field samples from naturally infected horses in Canada and Germany. Can J Vet Res. 2015;(79):161–9.
  15. Namdari S, Nicholson T, Abboud J, Lazarus M, Ramsey ML, Williams G, et al. Comparative study of cultures and next-generation sequencing in the diagnosis of shoulder prosthetic joint infections. J Shoulder Elbow Surg. 2019 Jan;28(1):1–8.

Reviewer 3 Report

Line 78: Samples were cryopreserved immediately after their extraction. What kind of semen extender was used to freeze the semen samples? Were spermatozoa separated from seminal plasma? Line 83: I think the environmental conditions were not the same as the ones for the first batch after 6 months. there are not seasonal variations in seminal bacterial composition? Lines 76,77: What kind of sperm quality tests were performed prior to seminal extraction? 172-178: Mändar have already pointed out that the presence of Prevotellaceae family alone or combined with Porphyromonadaceae are associated with a higher rate of reproductive inflammatory conditions. Our results may contradict this finding, as they are the predominant group in thenon-carrier state, especially Porphyromonadaceae. These conflicting findings can not due to the limitations of a single case análisis?

Author Response

Line 78: Samples were cryopreserved immediately after their extraction. What kind of semen extender was used to freeze the semen samples? We did not use any semen extender. We took a 1-1,5 ml semen aliquot and introduced it in liquid nitrogen. The frozen process had the only objective of preserving the microflora, but we did not need the spermatozoa to be alive. The rest of the semen was then used by the breeding centre for doses.

Were spermatozoa separated from seminal plasma? No, they were not. We took the aliquot after homogenizing the semen. We wanted to pick a semen sample as representative as possible.

Line 83: I think the environmental conditions were not the same as the ones for the first batch after 6 months. there are not seasonal variations in seminal bacterial composition? That is right. Batches were collected in different seasons, which could create a potential bias. However, we have also been working on seminal microbiome seasonality and, to date, we have observed no flora composition differences among seasons (not published data). Besides, we also found a study (1) that supports the idea of no flora variation among seasons in stallions.

Lines 76,77: What kind of sperm quality tests were performed prior to seminal extraction? We performed five different tests: total volume, gel-free volume, sperm concentration (spectrophotometry), total motility (computer-assisted sperm analysis) and progressive motility (computer-assisted sperm analysis). Information now included in lines 83-84.

172-178: Mändar have already pointed out that the presence of Prevotellaceae family alone or combined with Porphyromonadaceae are associated with a higher rate of reproductive inflammatory conditions. Our results may contradict this finding, as they are the predominant group in the non-carrier state, especially Porphyromonadaceae. These conflicting findings cannot due to the limitations of a single case analysis? Yes, you are right. Single-case studies do not allow drawing general conclusions. Our study is just a preliminary approach to a potential association between seminal flora variations and reproductive conditions. We are currently working on a case-control study with more samples, and we are observing flora variations between CEM-free and CEM-carrier animals (not published data). For sure, analysing just one sample is a limitation of our study and, thus, we have reflected it in the Discussion and Conclusions section (lines 269-270 and 275).

REFERENCE:

  1. Cerny KL, Little TV, Coleman RJ, Ball BA, Troedsson MH, Squires EL. Variations of Potentially Pathogenic Bacteria Found on the External Genitalia of Stallions During the Breeding Season. JEVS. 2015 Feb;35(2):170–3.

Reviewer 4 Report

Simple Summary:

In the simple summary section line 25, authors mentioned single-case study. Normally this means one animal was used. Here 6 and 14 stallions were used for the first and second phase of study respectively. This single-case can be rephrased.

Abstract:

In abstract it is written that main concern was detection of carriers in apparent agent free farms; line 29 and 30. The aim of the study was to compare differences in the seminal microflora in carrier and non-carrier state, line 33. In this context are the authors saying that NGS analysis of seminal microflora can be a tool to detect the carriers? Because the problem and the aim are not in right harmony. If they can rewrite that will make the manuscript better.

In line 42 again “single-case” analysis term was used.  

Introduction:

Introduction is too short. Again, here in line 61 and 62, aim of the study was repeated. It is also not clarified that seminal microflora study might be helpful to create an easy method to identify the carriers. This can be written in the manuscript.

Material and methods:

In first batch semen was collected from 6 stallions, line 71 and 72. Age, breed was not mentioned. Second batch was composed of 14 samples, line 82. It is not clear whether this 14 samples were from 14 different stallions. If authors can clarify this that will be good.

Results:

In Table 1. Line 111 mentioned “theoretical composition”. Does this composition have some journal article or company standard that authors can cite?

In Figure1. among 6 stallions only one was TE+ and rest was TE-? And in Figure2. Highlighted one was TE- and rest 13 was TE+?

So TE+ stallion was a carrier and in 6 months he was TE- without any medication as he has mild symptoms and that can be cured naturally? No control stallion was there who was normal and not a carrier in both first and second batch. Without a control comparison is not very precise. If you can explain this in the manuscript that will be good to understand the situation.

Discussion:

Is line 151 and 152 said TE is present in digestive system of stallions? Is this natural to present?

Conclusion:

here my comment is more general. Are you considering NGS of seminal microflora as a tool for future detection of carrier stallions? As in the light of COVID-19 we all know how carriers with no symptom can be harmful. Or authors are suggesting any other test that can detect carriers.

Author Response

Simple Summary:

In the simple summary section line 25, authors mentioned single-case study. Normally this means one animal was used. Here 6 and 14 stallions were used for the first and second phase of study respectively. This single-case can be rephrased. Here you have a point. However, we would like to keep talking about a single-case study, as statistics and discussion exclusively focuses on one animal.

Abstract:

In abstract it is written that main concern was detection of carriers in apparent agent free farms; line 29 and 30. The aim of the study was to compare differences in the seminal microflora in carrier and non-carrier state, line 33. In this context are the authors saying that NGS analysis of seminal microflora can be a tool to detect the carriers? Because the problem and the aim are not in right harmony. If they can rewrite that will make the manuscript better. Our study mainly tries to explain some inconsistent official CEM results. Some animals have tested positive to CEM without an epidemiological explanation supporting it. Several authors have pointed to a low sensitivity of official culture-based methods [1–3].

Here, we want to propose a new explanation of these inconsistencies from a microbiological point of view. We have detected a significant seminal microflora shift in the same animal in its carrier and non-carrier state. Therefore, the method we used to detect these states is not the focus of our study. For sure, NGS have some advantages comparing to culture (that we would cover by the end of this letter). But nowadays it is not a practical option for CEM diagnosis.

For sure, if you keep considering that lines 29-30 misleads readers, we can remove them from the paper.

In line 42 again “single-case” analysis term was used. Yes. As we previously said, we would prefer to maintain this expression, because we elaborated the discussion around just one animal.

Introduction:

Introduction is too short. We have added a new paragraph describing the contagious equine metritis infection (lines 52-56). Again, here in line 61 and 62, aim of the study was repeated. It is also not clarified that seminal microflora study might be helpful to create an easy method to identify the carriers. This can be written in the manuscript. Probably, in the near future, it would be plausible to analyze seminal microflora to identify CEM carriers with high-throughput sequencing. Despite, the main objective of our study is to give a new insight into natural seminal flora and Taylorella equigenitalis interactions, not to present a new CEM diagnostic method.

Material and methods:

In first batch semen was collected from 6 stallions, line 71 and 72. We have added age ranges of the first and second batch in lines 78 and 90, respectively. Second batch was composed of 14 samples, line 82. It is not clear whether this 14 samples were from 14 different stallions. If authors can clarify this that will be good. OK. Yes, batch 2 includes all animals from the first batch. Clarification included in lines 89-90.

Results:

In Table 1. Line 111 mentioned “theoretical composition”. Does this composition have some journal article or company standard that authors can cite? Yes, we used a commercial standard (ZymoBIOMICS Microbial Community Standard®, Zymo Research, CA). Information now included in lines 94-96. We have omitted this table at the suggestion of another reviewer. For sure, if you consider the inclusion of this Table is important for a good comprehension of the manuscript, we will include it again.

In Figure1. among 6 stallions only one was TE+ and rest was TE-? Yes, only one animal (subject of the study) was positive to Taylorella equigenitalis. It is represented with a big yellow ball. And in Figure2. Highlighted one was TE- and rest 13 was TE+? In Figure 2, there were no positives to Taylorella equigenitalis. The big grey ball represents the animal from the first batch that was positive. What we wanted to represent in both figures was a dissimilarity test among individuals. Figure 1 shows that the infected animal (first batch, big yellow ball) had a quite different seminal microflora comparing to non-infected animals, as it stands far from the rest of samples in the graph. However, in Figure 2 (second batch, big grey ball), when Taylorella is not detected in the animal anymore, it belongs to the general group of samples.

So TE+ stallion was a carrier and in 6 months he was TE- without any medication as he has mild symptoms and that can be cured naturally? No control stallion was there who was normal and not a carrier in both first and second batch. Without a control comparison is not very precise. If you can explain this in the manuscript that will be good to understand the situation. Here you have a point. Unfortunately, Taylorella equigenitalis produce no symptoms in stallions, just in mares. This made impossible to conduct a case-control study, which would have provided a much better design.

Discussion:

Is line 151 and 152 said TE is present in digestive system of stallions? Is this natural to present? No, Taylorella is not present in the digestive system. We have changed the sentence construction, as it can create confusion (lines 188-191).

Conclusion:

here my comment is more general. Are you considering NGS of seminal microflora as a tool for future detection of carrier stallions? As in the light of COVID-19 we all know how carriers with no symptom can be harmful. Or authors are suggesting any other test that can detect carriers. It is true that NGS is a much more sensitive technique analyzing microbial flora composition than CEM official techniques as culture. For example, NGS is not limited by genera fastidious to cultivate [4] and it overcomes limitations of culture as contamination. Despite of that, NGS also has limitations, for example, differential amplification of primers [5], bias in DNA extraction or underdevelopment of high-quality interpretation of data [6]. Moreover, despite NGS has proven to be more cost-effective [7] than other techniques, this only applies with a large number of samples. Routine testing of farms do not get to that level of samples, which will make NGS testing unaffordable.

With our preliminary study we just wanted to shed new light on a possible therapeutic approach of CEM carriers. As it happened in the digestive system, a better understanding of the role of some bacteria in the host can open the door to new pre and probiotic therapies [8,9]. For sure, there is a much work to be done before having new, on-field therapies. We hope that our work may start a new line of research in this area.

REFERENCES:

  1. Kristula, M.; Smith, B. Diagnosis and treatment of four stallions, carriers of the contagious metritis organism--case report. Theriogenology 2004, 15, 2–3, doi:10.1016/s0093-691x(03)00228-0.
  2. Schulman, M.L.; May, C.E.; Keys, B.; Guthrie, A.J. Contagious equine metritis: Artificial reproduction changes the epidemiologic paradigm. Veterinary Microbiology 2013, 167, 2–8, doi:10.1016/j.vetmic.2012.12.021.
  3. Timoney, P.J. Horse species symposium: contagious equine metritis: an insidious threat to the horse breeding industry in the United States. J. Anim. Sci. 2011, 89, 1552–1560, doi:10.2527/jas.2010-3368.
  4. Namdari, S.; Nicholson, T.; Abboud, J.; Lazarus, M.; Ramsey, M.L.; Williams, G.; Parvizi, J. Comparative study of cultures and next-generation sequencing in the diagnosis of shoulder prosthetic joint infections. J Shoulder Elbow Surg 2019, 28, 1–8, doi:10.1016/j.jse.2018.08.048.
  5. Lambert, J.A.; Kalra, A.; Dodge, C.T.; John, S.; Sobel, J.D.; Akins, R.A. Novel PCR-Based Methods Enhance Characterization of Vaginal Microbiota in a Bacterial Vaginosis Patient before and after Treatment. Appl Environ Microbiol 2013, 79, 4181–4185, doi:10.1128/AEM.01160-13.
  6. Mändar, R.; Punab, M.; Korrovits, P.; Türk, S.; Ausmees, K.; Lapp, E.; Preem, J.-K.; Oopkaup, K.; Salumets, A.; Truu, J. Seminal microbiome in men with and without prostatitis. Int. J. Urol. 2017, 24, 211–216, doi:10.1111/iju.13286.
  7. Torchia, M.T.; Austin, D.C.; Kunkel, S.T.; Dwyer, K.W.; Moschetti, W.E. Next-Generation Sequencing vs Culture-Based Methods for Diagnosing Periprosthetic Joint Infection After Total Knee Arthroplasty: A Cost-Effectiveness Analysis. J Arthroplasty 2019, 34, 1333–1341, doi:10.1016/j.arth.2019.03.029.
  8. Clark, A.; Sallé, G.; Ballan, V.; Reigner, F.; Meynadier, A.; Cortet, J.; Koch, C.; Riou, M.; Blanchard, A.; Mach, N. Strongyle Infection and Gut Microbiota: Profiling of Resistant and Susceptible Horses Over a Grazing Season. Front Physiol 2018, 9, 272, doi:10.3389/fphys.2018.00272.
  9. Morrison, P.K.; Newbold, C.J.; Jones, E.; Worgan, H.J.; Grove-White, D.H.; Dugdale, A.H.; Barfoot, C.; Harris, P.A.; Argo, C.M. The Equine Gastrointestinal Microbiome: Impacts of Age and Obesity. Front Microbiol 2018, 9, 3017, doi:10.3389/fmicb.2018.03017.

Round 2

Reviewer 1 Report

The manuscript has been greatly improved and its appreciated that you have considered previous suggestions. I now consider that it is apt for publication with one suggestion. Table 3 should be removed and substituted by a bar graph depicting mean relative abundance (%) of samples in batch one (without TE+ sample), relative abundance of TE+ sample, mean relative abundance of samples in batch two (without TE- sample) and relative abundance of TE- sample. This way it would be more visible the fact that TE+ sample has different phylum composition in batch one than TE- in batch two.

Author Response

The manuscript has been greatly improved and its appreciated that you have considered previous suggestions. I now consider that it is apt for publication with one suggestion. Table 3 should be removed and substituted by a bar graph depicting mean relative abundance (%) of samples in batch one (without TE+ sample), relative abundance of TE+ sample, mean relative abundance of samples in batch two (without TE- sample) and relative abundance of TE- sample. This way it would be more visible the fact that TE+ sample has different phylum composition in batch one than TE- in batch two.

We have removed Table 3 and added a new figure according to your suggestions (lines 161-165).

We would like to thank you very much for all your valuable comments and suggestions, which have helped us to improve the quality of our manuscript.

Reviewer 2 Report

Dear authors,
Thank you for following my guidelines.

The manuscript has been improved and is suitable to publish in Animals in the present form.

Author Response

Dear authors,

Thank you for following my guidelines.

The manuscript has been improved and is suitable to publish in Animals in the present form.

We thank you for your careful reading of our manuscript and your insightful comments and suggestions.

Reviewer 4 Report

N/A

Author Response

N/A

We would like to thank you for your thoughtful comments and efforts towards improving our manuscript.

This manuscript is a resubmission of an earlier submission. The following is a list of the peer review reports and author responses from that submission.

Round 1

Reviewer 1 Report

Comments to the authors:

Study describing the seminal microbiota determined by next-generation sequencing from one stallion in two conditions: presence or absence of Taylorella equigenitalis.

This study has several potential scientific interests due to the scarce description of seminal microbiota in the equine species and the association with an infectious agent important in fertility reduction in mares due to development of contagious equine metritis. However, I have a few concerns regarding the study design description, results presentation and discussion/conclusions drawn. Specific comments are detailed below:

SUMMARY AND ABSTRACT

It is not clear how the study was really conducted. Was the stallion naturally infected by Taylorella equigenitalis? How was this infection determined? These concerns are derived from the materials and methods description, if that is improved it needs to be reflected as well in the abstract. As well, limitations of this study should be as well briefly discussed in the abstract.

INTRODUCTION

Line 49: Taylorella equigenitalis should be written in italics

Line 59: Include word “in” (seminal flora composition may have an impact “in” venereal agent survival).

MATERIALS AND METHODS

This section needs to be improved since is not clear how the study was really designed. You comment that originally six semen samples were analyzed, but it is not clear if they come from the same stallion or if it is one sample from 6 different stallions. Later on, at the results section, you describe that 14 samples were evaluated. Again this is not clear where they come from. Please describe all the details about sample collection, at which times and from which animals were collected and why. Was a sample size calculated originally?

It is not clear how the stallion became infected from Taylorella equigenitalis, naturally or experimentally. How was really determined that T. equigenitalis was present, by seminal culture, PCR or only by next-generation sequencing? Was the first sample from this stallion positive and the second one negative, from the abstract and title it looks like first sample was negative and second positive, but from the rest of manuscript it looks the opposite? Why did the stallion clear off the infection?

Regarding the data analysis, further description is needed on how beta diversity and taxonomic levels were calculated. If beta diversity is reported is because there are several samples from different stallions. However, the main focus of the materials and methods is described only in one stallion (n=1). Explain better how all this analysis comprises all the samples from different stallions and which analysis are performed only with the sample from the infected stallion. Other bioinformatic analysis could be included such as alpha diversity.

Was any quality control included? DNA concentration determination after DNA extraction? Specific PCR V3 amplification? Mock community inclusion as sequencing control?

RESULTS

Lines 93-95 belong to materials and methods or discussion, not results.

This section needs reorganization and more data description. Divide results describing first the results of analysis including all the stallion samples excluding the infected stallion (from M&M it is not clear where figure 1 and 2 come from). Include taxonomic family level graphs (so it can be compared to the infected stallion). It would be interesting to show if there is individual variation in microbiota composition when the same stallion is retested, and in which families this variation is seen.

Secondly, include a separate section on how one stallion was selected because of T. equigenitallis infection and the results of its microbiota analysis.

DISCUSSION

This section needs further development.

Table 2 belongs to results section, not to discussion. In this table phylum and families are mixed and it is not clear. Please clarify the taxonomy of what is represented to the same level.

A description of the limitations of this study needs to be performed. Very importantly, it needs to be discussed that the n=1 limitation precludes the authors to draw the conclusion that what they observed can be extrapolated to general population. “the Corynebacteriaceae family predominant group, representing  the contagious equine metritis carriers; and the Bacteroidetes phylum predominant group for the non-carrier stallions” this conclusion cannot be drawn from the results, it is true only for the stallion studied, but not for all non-carrier stallions. A higher sample size would be needed to report such a conclusion.

Reviewer 2 Report

This manuscript studied the microbiome in semen of ine stallion before and after an infection with Taylorella equigenitalis. Semen was collected from only one stallion, and there were no controls. The materials and methods are poorly described. Six semen samples were studied with next generation sequencing. It is not described how these samples were collected and how often. More importantly, how contamination of the samples by microbes in the environment, or in any of the materials that come into contact with the samples before analyses, was avoided was not described. The manuscript cannot be accepted for publication.

Reviewer 3 Report

In the manuscript under the title:"Comparative semen microbiota composition pre and post Taylorella equigenitalis infection", authors compare the differences in the seminal microbiome composition of one stallion before and after infection by Taylorella equigenitalis, with the use ion torrent sequencing.

In my opinion, the manuscript in present form is not acceptable for publication.

Introduction.

The Taylorella equigenitalis infection in equines should be far better described.

ln 50: "if not treated properly"- What does it mean?

ln 56-56: The sentence is not clear at all. The hypothesis and aim of the study are not clearly depicted.

Materials and methods.

It seems that is not a research study but rather a case report. 

The description of sequencing is poor.

results and discussion do not support a better understanding of survival mechanisms of the bacteria.